# Scenario-Based Analysis of Land Use Competition and Sustainable Land Development in Zhangye of the Heihe River Basin, China

**DOI:** 10.3390/ijerph181910501

**Published:** 2021-10-06

**Authors:** Yuping Bai, Zhe Zhao, Chuyao Weng, Wenxuan Wang, Yecui Hu

**Affiliations:** 1School of Land Science and Technology, China University of Geosciences, Beijing 100083, China; baiyp@cugb.edu.cn (Y.B.); wengcy@cugb.edu.cn (C.W.); 2Institute of Geographic Sciences and Natural Resources Research, Chinese Academy of Sciences, Beijing 100101, China; 201401021124@sdust.edu.cn; 3School of Economics, Liaoning University, Shenyang 110136, China; zhaozlnu@163.com; 4College of Resources and Environment, University of Chinese Academy of Sciences, Beijing 100101, China

**Keywords:** land use competition, scenario simulation, CGELUC model, DLS model, Zhangye city

## Abstract

Rapid economic growth has a significant impact on land use change, which would threaten the natural ecology. Zhangye city of the Heihe River Basin, China is an ecologically vulnerable region where land use changes significantly due to socioeconomic development and population increases. The study employed a computable general equilibrium of land use change (CGELUC) model to simulate land use change and then used a dynamic land system (DLS) model to spatialize land use change during 2015–2030 under three development scenarios in Zhangye city. The three development scenarios are the baseline scenario (BAU), the resource consumption scenario (RCS) and the green development scenario (GDS). We found that economic growth would lead to land demand increases in high value-added industries and decreases in low value-added industries. The cultivated land would decrease while the built-up area would increase. By 2030, the cultivated land will decrease by 8.16%, 10.89% and 4.16%, respectively, under BAU, RCS and GDS, while the built-up area will increase by 8.61%, 10.39% and 4.75%, respectively. The expansion of built-up area under RCS presents spatial characteristics of centralized distribution, while spatial characteristics of uniform discrete distributions are presented under GDS. The expansion of ecological land under GDS would be considerable, especially in the north of Sunan County and Gaotai County, and around the natural reserve of Ganzhou County. This paper provides a scientific reference for coordinating economic development and ecological protection in the rapidly developing urbanized areas in western China.

## 1. Introduction

Land use change has both social and natural attributes, and is influenced by the combination of climate change and human activities [1]. Since natural factors are relatively stable with cumulative effects over short time scales, human activities have played a dominant role in regional environmental change at the scale of decades to centuries [2]. With the rapid development of social economy, the role of human socioeconomic activities in driving the evolution of land use systems has become more obvious [3,4]. Since the reform and opening up, China has experienced rapid urbanization [5]. The rapid growth of urbanization has had a significant impact on land use change. The construction of socioeconomic development scenarios to analyze the impact of human activities on land use change has indicated the direction for land resource management in the context of sustainable development.

The Heihe River Basin is the second largest inland river basin in northwest China, located in the middle of the Hexi Corridor, China. The midstream area of Heihe River Basin accounts for 88.47% of the population size and 87.93% of the GDP of the whole basin [6]. As a typical city in the middle reaches of the Heihe River Basin, Zhangye city is still in an underdeveloped stage [7,8] However, in recent years, with the implementation of the “Silk Road Economic Belt” strategy, the People’s Government of Zhangye Municipality proposed to build Zhangye into an important node city of the “Silk Road Economic Belt” [9,10]. In this context, the intensity of human socioeconomic activity will be inevitably greater than that of other cities in Heihe River Basin, and the land and water resource systems will be more severely affected by human socioeconomic activity in this area [11]. Therefore, it is very typical to take Zhangye as an example to analyze the land use competition of the Heihe River Basin.

In the accelerating process of urbanization, the conflict between land supply and demand is becoming increasingly prominent [12]. Optimized allocation of land use has become an effective way to realize the sustainable use and management of land resources [13,14]. Zhangye is a typical traditional agricultural city, known as “Golden Zhangye”, but the industrial structure is at a low level. The strategic concept of the “Silk Road Economic Belt” creates opportunities for the transformation and urbanization development of Zhangye city, which will inevitably lead to a major change in land use types. On the other hand, the development and utilization of land resources has a great impact on the ecological environment quality of the Zhangye Oasis. Ecological security is threatened. Ecological restoration, such as returning farmland, closing hillsides for afforestation and wetland protection, is particularly urgent. The optimal allocation of land use, especially the optimization of land use spatial patterns, is of great significance for improving land use efficiency and maintaining the balance of land ecosystem [15].

The optimal allocation of land use is an important way to realize sustainable land use and promote rapid the development of the regional economy and a harmonious environment [16,17]. Since the 1970s, in order to meet the needs of the sustainable development of land use, research on the theory and methodology of optimal allocation of land use has been deepening. The optimization of land use allocation based on model construction is common in current studies. The models of the optimal allocation of land use structure mainly include the linear programming method [18,19,20], the multi-objective linear programming method [21,22,23], the system dynamics model [24,25,26,27] and the computable general equilibrium model [28,29,30]. The weakness of the linear programming method is that the objective is relatively simple and the objective equation may have no solution. In addition, the linear programming method can only analyze the allocation of land resources statically. Wang et al. [21] established an uncertain interval multi-objective linear programming model to assess the suitability of land use structure in Pi County of Sichuan Province. However, the difficulty of multi-objective linear programming is how to quantify each benefit goal. Luo et al. [31] combined a system dynamic (SD) model and a CLUE-S model to analyze land use dynamics at different scales in Sangong watershed in Xinjiang, China. Based on the SD model, He et al. [32] analyzed the carrying capacity of cultivated land in Bijie City, Guizhou Province, and simulated the changes of cultivated land under three different scenarios of population control-oriented, economic-development-oriented and coordinated development. However, there are still some theoretical deficiencies in how to determine parameters when carrying out long-term projection based on SD models. Compared with other models, the computable general equilibrium (CGE) model is a multi-sector and non-linear microscopic mechanism model that could include resource constraints. By establishing a CGE model embedding land resource accounts, the impact of social and economic systems on land use change can be analyzed accurately [33,34,35].

Land use change, including the fluctuation of planting area of food crops and cash crops, as well as the changes in land use types such as forest, grassland and wetland, are closely related to resources, environment and economic activities. Therefore, the CGE model related to land use change can be used to study the influence of different types of land use on the economic system, especially for the influence of agricultural-related sectors, as well as the influence of other sectors, trade and the environment in the economic system. Previous studies focus on issues such as wetland conservation programs, land erosion and degradation, agricultural land and forestry conservation policies and climate change policies related to land use change, using a CGE model embedding land resource accounts (Table 1). However, currently, many CGE models do not consider the land supply constraint caused by the heterogeneity of different land use types. Therefore, they cannot simulate land conversion among different uses, and the land supply response to price change would be exaggerated. The characterization of land heterogeneity is an important but challenging issue in CGE modeling.

Many models are employed to simulate the spatial distribution of land use, including the CLUE-S model [42,43,44], the agent-based model [45,46,47], the cellular automata (CA) model and the dynamic of land system (DLS) model [41,48]. Zhang et al. [49] adopted the models of CLUE-S and SWAT to simulate the pollution load under land use scenarios in the upstream basin of the Miyun Reservoir in Beijing, China. Villamor et al. [50] used an agent-based model to explore the trade-offs between the provision of ecosystem services and rural income under a series of land use dynamic scenarios in lowland Sumatra. Guan et al. [51] adopted a method of CA and the Markov model to simulate the spatial and temporal distribution of land use in Saga City from 2015 to 2042. Compared with the traditional models, the DLS model constructs a spatial statistical model of land system spatial distribution and its driving factors in a spatially explicit way, which can better reflect the complex self-organization and the competition process among land use types in a land system [52,53]. Based on the principle of spatial allocation and the balance of land supply and demand, the DLS model considers the driving mechanism of natural factors and social–economic factors on land use change, and realizes the dynamic simulation of a land system on a fine grid scale. Under the integration of the CGE and DLS models, the dynamic macro-analysis of land systems is closely combined with micro-econometric analyses on the grid scale. The comprehensive consideration of regional socioeconomic development characteristics, natural conditions, policy conditions and historical trends of land use change is realized to make the simulation results more scientific and reasonable.

Therefore, this paper developed a computable general equilibrium of land use change (CGELUC) model to simulate land use change in Zhangye city of Heihe River Basin, China in 2015–2030 under three development scenarios: baseline, resource consumption, and green development. The dynamic land system (DLS) model was further applied to spatialize land use change on the spatial grid scale. This paper aims to provide policy decision information on regional land resource allocation and management, regional economic sustainable development, and ecological environment protection in rapid urbanization areas in western China.

## 2. Study Area

Zhangye city of Heihe River Basin, China extends 97°20′–102°12′ E and 37°28′–39°57′ N, and is located at the northwest of Gansu Province in the middle section of the Hexi Corridor and the middle and upper reaches of Heihe River, the second largest inland river in China. The cultivated land covers an area of 4204 km^2^, and still about 2000 km^2^ land is available for reclamation. There are 26 rivers distributed in this region, with an annual runoff of 2.66 billion m^3^. The groundwater resources are also abundant. This area is characterized by an arid continental climate, the mean annual temperature is 6 °C, and the annual sunshine is 3000 h.

Currently, Zhangye has 6 counties, including Linze county, Gaotai county, Shandan county, Minle county, Sunan county and Ganzhou district (Figure 1). According to the statistical data, the GDP reached CNY 37.70 billion in 2017, which is nearly 3.4 times higher than that in 2005.The industrial structure also shows obvious characteristics of an increasing trend in the proportion of industry and the service industry. With the continuous optimization and adjustment of the industrial structure, the proportion of the service industry shows an increasing trend, reaching 54.75% in 2017 (Figure 2).

The land use change in Zhangye from 1990 to 2015 indicates that forest area, grassland, water area and unused land showed a decreasing trend, while cultivated land and built-up area showed an increasing trend (Table 2). With the continuous acceleration of the urbanization process, the demand for built-up area is large. However, the areas suitable for construction are distributed in high-quality cultivated land, and the contradiction between major infrastructure construction and cultivated land protection is prominent. On the other hand, Zhangye city is located in the intersection of the Qinghai Tibet Plateau and the Loess Plateau, belonging to the western ecological security barrier. However, in recent years, with a large area of cropland having been reclaimed, cropland and forest compete for water, leading to more dying ecological forests and aggravated desertification.

## 3. Methodology and Data

This paper developed a computable general equilibrium of land use change (CGELUC) model to simulate the changes of land use structure in Zhangye city of Heihe River Basin, China, in 2015–2030 under three development scenarios: baseline, resource consumption, and green development. The dynamic land system (DLS) model was further applied to spatialize land use change on a spatial grid scale. The research framework is shown as follows (Figure 3):

### 3.1. CGELUC Model

In this study, we developed a computable general equilibrium of land use change (CGELUC) model to simulate the changes in land use structure by using the ORANI-G model, which is a single-country CGE model and is widely applied to the analysis of economic policies and academic research around the world (https://www.copsmodels.com/oranig.htm) (Accessed on 1 August 2018) [35]. In this study, we expanded the ORANI-G model by embedding a land resources account. Similar to the general CGE model, this model includes production, consumption, trade, tax, income distribution, markets and other modules. The dynamic recursive mechanism is established in the model, which mainly aid in scenario prediction and the analysis of future land system structure dynamics. The model uses Gempack software (Centre of Policy Studies, Victoria University, Melbourne, Australia) as the model running environment.

#### 3.1.1. Production Module

The following figure describes the structure of the production module of the CGELUC model. The module encompasses multi-level nested production functions (Figure 4), each describing an optimal combination to ensure the minimization of production costs. The top-level is the optimal combinations of goods, production factors, and other costs using Leontief functions. The second level is the combination of domestic and imported commodities using the CES function. Land resources are embedded in the production module as a production factor and nested with other production factors, labor and capital, by the CES production function.
(1)$$LND_{i}=slnd_{i}\cdot {\left[\frac{PPRIM_{i}}{PLND_{i}}\right]}^{\sigma prim_{i}}$$
(2)$$LAB_{i}=slab_{i}\cdot {\left[\frac{PPRIM_{i}}{PLAB_{i}}\right]}^{\sigma prim_{i}}$$
(3)$$CAP_{i}=scap_{i}\cdot {\left[\frac{PPRIM_{i}}{PCAP_{i}}\right]}^{\sigma prim_{i}}$$
(4)$$PPRIM_{i}={\left[slnd_{i}\cdot PLND_{i}{}^{1-\sigma prim_{i}}+slab_{i}\cdot PLAB_{i}{}^{1-\sigma prim_{i}}+scap_{i}\cdot PCAP^{1-\sigma prim_{i}}\right]}^{\frac{1}{1-\sigma prim_{i}}}$$

Considering the heterogeneity of land resources and the constraints of land conversion, total land input is divided into different uses according to conversion elasticity and relative rent by applying the CET function. In this study, we adopted the method proposed by Horridge [54] to amend the CET function in order to achieve the allocation of areas of different land use types under the total fixed land area, which avoids the errors in the traditional CGE model when it is used to explain changes in physical units.
(5)$$LND_{i}=LNDs\cdot {\left[\frac{PLND_{i}}{PLNDs}\right]}^{\sigma }$$
(6)$$PLNDs=\sum_{i}S_{i}*PLND_{i}$$
(7)$$QLND_{i}=LND_{i}{}^{\alpha_{i}}$$
(8)$$QLNDs=\sum_{i}QLND_{i}$$

Description of variables in equations are listed in Appendix A.

#### 3.1.2. Construction of Dynamic Mechanism

The static CGE model describes the behavior and characteristics of the main economic system in a given period. In this study, the model introduced a dynamic module to simulate the impact of socioeconomic factors on land use change under different future scenarios. The model introduces recursive dynamic structures of capital accumulation and employment to depict the evolution of the economic system, in order to track the trend of economic change over time under continuous policy and economic shocks and predict dynamic long-run macroeconomic change. Long-term forecasting is carried out on the basis of the dynamic accumulation of capital, which affects the total capital of the next round of production equal to the capital stock of the previous period minus depreciation plus total investment. In addition, the model takes future changes of labor employment into account, constructing a dynamic adjustment mechanism for wages and employment. 

Other modules and specific explanations can be found in Bai et al. [41].

### 3.2. DLS Model

#### 3.2.1. The Structure of DLS

The dynamic of land system (DLS) model is based on the theory of equilibrium in regional land use structure change and the constraint of land use type distribution at the grid scale. Based on a comprehensive consideration of the driving mechanism of natural factors and socioeconomic factors in land use changes, a dynamic simulation of spatial pattern evolution in regional land use change is realized.

The DLS model consists of the following modules:

(1) Driving analysis module of land use distribution—this module is used to measure the spatial relationship between natural environmental factors, climatic factors, location factors, socioeconomic factors and land use distribution, and depict the control and transformation effect of these driving factors on land use;

(2) Spatial allocation module—based on the probability of land use conversion, land use conversion rules and land use conversion area restrictions, the spatial allocation of land use change is effectively simulated through the land supply and demand balance at the grid scale.

#### 3.2.2. Construction of DLS Model

The DLS model fully considers the driving characteristics of regional land use type distribution and the characteristics of land supply and demand balance, reflecting the systems, mechanisms, equilibrium and convergence of regional land systems.

The changes in land supply and demand in the DLS model are a dynamic convergent process in which the total amount remains in equilibrium. The raster-scale land supply and demand balance requires a highly accurate raster-scale land supply and demand balance model, which is an empirical statistical model with spatially explicit typical characteristics. By considering the effects of local drivers within each raster and the neighborhood effects of drivers around the raster, the competition process between land use types is finely depicted. The core is the process of the spatial allocation of land demand at the micro-scale (Figure 5).

It is assumed that the probability of land use type k occurring, based on the local raster factor i, is $p_{i,Local}^{k}$, based on the neighborhood raster factor i being $p_{i,NBH}^{k}$. The combination probability of land use, $p_{i}^{k}=f\left(p_{i,Local}^{k},p_{i,NBH}^{k}\right)$, reflects the self-organization process of land use system evolution. $p_{i,Local}^{k}$ and $p_{i,NBH}^{k}$ are obtained from the nonlinear constraint model of land use type distribution at the raster scale:(9)$$\begin{array}{cc} \ln\left(\frac{p_{i}^{k}}{1-p_{i}^{k}}\right) & =a_{0}^{k}+a_{1}^{k}x_{i1}+a_{2}^{k}x_{i2}+\cdots +a_{l}^{k}x_{il}+\cdots a_{L}^{k}x_{iL}+r{\hat{y}}_{i}^{k}\\ & =a_{0}^{k}+a_{i}^{k}X_{i}+r{\hat{y}}_{i}^{k} \end{array}$$
where $p_{i}^{k}$ is the probability of the raster i displaying the land use type of k, and $X_{il}$ is the driving factor l in the raster i, which can be a local factor or neighborhood factor. $a_{i}^{k}$ is the regression coefficient and r is the influence coefficient of the spatial autocorrelation factor. The spatial autocorrelation factor ${\hat{y}}_{i}^{k}$ is also defined.

### 3.3. Data

#### 3.3.1. Socioeconomic Data

This paper developed a CGELUC model to study the impact of a socioeconomic system on land use change under different scenarios in Zhangye city from 2015 to 2030. The model used the Zhangye Input–Output Table 2012, obtained from the Bureau of Statistics of Zhangye, China [55], as the basis, which includes 7 agricultural sectors and 41 secondary and tertiary industrial sectors, shown in Appendix B. At the same time, it is necessary to calculate the value of land resources in Zhangye city. This paper mainly introduces the division of land resources and the accounting methods of land resource value in 48 sectors.

#### 3.3.2. Natural Data

Land use data and other natural environment data, including landform type, DEM, slope, aspect, etc., are provided by the Resource and Environment Science and Data Center, Chinese Academy of Science (http://www.resdc.cn/dataResource/dataResource.asp) (accessed on 1 August 2018). Climate data, including temperature, precipitation, sunshine duration (sun), ≥0 °C accumulated temperature, and ≥10 °C accumulated temperature, are provided by the Chinese Meteorological Data Center (http://data.cma.cn/) (accessed on 1 August 2018). Soil data including soil organic, soil depth and soil PH value are provided by the Harmonized World Soil Database (HWSD). Social economic data such as population and GDP at 1 km*1 km resolution and date related to distance are also derived from the Resource and Environment Science and Data Center, Chinese Academy of Science (http://www.resdc.cn/dataResource/dataResource.asp) (accessed on 1 August 2018).

#### 3.3.3. Value Accounting of Land Resources

The input–output table of 48 sectors in Zhangye city in 2012 is a value input–output table. To embed the land resources account into it, it is necessary to carry out the value accounting of land resources. Referring to the System of Integrated Environmental and Economic Accounting (SEEA), the value of agricultural land is multiplied by land rent, and the value of industrial and service land is multiplied by land transfer price divided by service life.

Based on the sectoral classification of the input and output tables of 48 sectors in Zhangye city in 2012, the land areas of seven agricultural sectors, including corn, wheat, oil crops, vegetables, cotton, fruit and other agriculture, were obtained, and the land price was calculated according to the average contracted price of local agricultural land, which is CNY 0.75/m^2^. Through remote sensing interpretation technology combined with field investigation, the areas of land functional types of 41 secondary and tertiary industrial sectors were determined, and the land prices were calculated according to the average price of the local land auction divided by the service life, which are respectively CNY 56.85/m^2^ and CNY 13.66/m^2^.

### 3.4. Scenario Design

There are many factors influencing the change of land use demand in social economic system. In national “Silk Road” economic belt planning in China, as the core zone of the Hexi Corridor, the rapid development of urbanization in Zhangye city is inevitable. Regional population, economic aggregate, industrial structure, consumption pattern, technology and management level all affect the land use demand of social and economic system. With the growth of the economy and population, the improvement of urbanization level and the transformation of consumption pattern, the demand for land use in social and economic systems would continue to rise, while the optimization and transformation of economic structure, the improvement of scientific and technological level and the progress of management measures would reduce the land use demand and improve the efficiency of land resource use. Therefore, it is of great practical significance to clarify the change trend of driving factors of socioeconomic development scenarios and simulate the land use demand of socioeconomic systems for guiding sustainable land management in Zhangye. Based on the literature [56,57] and regional development planning, the projection of population, investment, consumption, economic growth and trade in Zhangye city in 2030 was carried out. Based on the CGELUC model, long-term closure with employment growth as the main factor was set. Meanwhile, the trend of future social and economic development is simulated according to the prediction index of future social and economic development (Table 3), and the baseline scenario (BAU) of future social and economic development is constructed.

This study simulates land use dynamics in Zhangye city of Heihe River Basin, China in 2015–2030 under three development scenarios: baseline (BAU), resource consumption (RCS), and green development (GDS). Referring to the BAU scenario, the trends of future social and economic development are also simulated in RCS and GDS scenarios (Table 3). The baseline scenario simulates the trend of future socioeconomic development and the demand for land use in the socioeconomic system with the same changes in investment, exports, resident consumption, capital input, population employment and labor technological progress from the past to the present. The resource consumption scenario gives priority to rapid economic growth and social development, and has a stronger impact on the demand for land use in the socioeconomic system by being shocked by higher investment, exports, residents’ consumption, capital, population employment and labor technological progress. The green development scenario makes more consideration of regional ecological environment construction, assuming that ecological security and ecological restoration will be mainly considered in future urban development. Compared with the basic scenario, investment, export, resident consumption and capital investment have a weaker impact on the demand for land use in the socioeconomic system, while the assumptions of population employment and labor technological progress remain unchanged.

## 4. Results

### 4.1. Land Use Change in Social–Economic System under Different Scenarios

According to the setting of the change rate of the shock variables, it is predicted that by 2030, the GDP under the baseline scenario will reach CNY 77 billion, with an annual growth rate of about 6%. This is consistent with regional development planning, showing that the scenario’s design is relatively reasonable.

This paper simulates the land demand of the socioeconomic system in Zhangye city from 2015 to 2030 under the three future scenarios. Figure 6 shows the percentage of land use change in different industries predicted up to 2030. The changes under the resource consumption scenario are the most intense, followed by the baseline scenario, and finally the green development scenario. From the perspective of industrial structure, the land use demands for wheat, corn and cotton in agricultural sectors decrease by more than 10%. The reason for this result is, on the one hand, with the development of modern agriculture, the efficiency of land resource utilization in traditional crop production is constantly improving, intensive management is strengthening, and the quality of cultivated land is improving. On the other hand, the internal structure of agricultural sectors in Zhangye city is optimized and adjusted. Agriculture has gradually changed from planations to the dual structure of grain and industrial crops, and even the triadic structure of grain, industrial crop and grass, which promotes a virtuous circle in the internal structure of agricultural sectors. In the secondary industry, the steady development of the social economy has stimulated the demand for land in mining, smelting and processing, food manufacturing, electric heat supply, construction and other industries. In particular, the demand for land in the general and special equipment manufacturing industry has increased by more than 50%. The land demand for most of the tertiary industries also showed an increasing trend, especially in the land demand of the real estate industry, which increases by more than 130% year-on-year. The land demands of other modern service industries, such as environment and public facilities management, information software technology, finance and health, social security and social welfare, are also stimulated by the socioeconomic system.

Figure 7 shows the percentage change of land demand for the three major industries under different scenarios in the future. From the perspective of the industrial structure, the land demand of the primary industry shows a downward trend on the whole, the land demand of the secondary and tertiary industries shows an increasing trend, and the land demand of the tertiary industry is higher than that of the secondary industry. The analysis shows that this is the result of the continuous optimization and upgrading of the industrial structure and the outward transfer of rural labor. The simulation results show that the land demand of the three major industries shows a development trend in the same direction under the three scenarios, but the change degree is different. The analysis shows that the continuous growth of investment, consumption and export will promote the rapid development of the economy; the increase in population employment and the technological progress of labor force will promote industrial transformation and upgrading; the economic structure will be adjusted and optimized, so that the land demand of the agricultural sector is reduced; the land demand of the secondary and tertiary industries will increase, and within a certain level of development, the greater impacts of the social and economic systems are affected by the coupling of investment, consumption, export, capital quantity, population employment and technological progress of labor force. The lower the land demand of the agricultural sector, the greater the land demand of the secondary and tertiary industries.

### 4.2. Comparative Analysis of Land Use Structure Changes under Different Scenarios

Based on the analysis of the changes in land demand in different industrial sectors under different future scenarios, 7 agricultural sectors and 41 industrial and service sectors are merged (Figure 8). It is found that the same land use types present the same direction variation trend under the three scenarios, but the change degree is different. The cultivated land decreases by 343 km^2^ (−8.16%), 453 km^2^ (−10.89%) and 175 km^2^ (−4.16%), respectively, under the BAU, RCS and GDS scenarios from 2015 to 2030 in Zhangye city. The built-up area increases by 29 km^2^ (8.61%), 35 km^2^ (10.39%) and 16 km^2^ (4.75%), respectively, under the BAU, RCS and GDS scenarios from 2015 to 2030. The forest land increases by 111 km^2^ (2.60%), 64 km^2^ (1.50%) and 582 km^2^ (3.81%), respectively, under the BAU, RCS and GDS scenarios from 2015 to 2030. The grassland increases by 384 km^2^ (2.58%), 224 km^2^ (1.51%) and 582 km^2^ (3.81%), respectively, under the BAU, RCS and GDS scenarios from 2015 to 2030. The water area increases by 18 km^2^ (2.62%) and 25 km^2^ (3.96%), respectively, under the BAU and GDS scenarios from 2015 to 2030, and decreases by 28 km^2^ (−10.39%) under the RCS scenarios from 2015 to 2030. The unused land decreases by 198 km^2^ (−1.31%) and 611 km^2^ (4.12%), respectively, under the BAU and GDS scenarios from 2015 to 2030, and increases by 117 km^2^ (0.77%) under the RCS scenario from 2015 to 2030.

### 4.3. Spatial Patterns of Land Use Change under Different Scenarios

The DLS model is further applied to simulate the land use spatial pattern of Zhangye city. Based on the land use data interpreted by remote sensing in 2015, the simulated results obtained are compared with the actual land use spatial pattern to test the accuracy of the DLS model. The consistency of the simulation results can be quantitatively reflected by using the Kappa index. In this study, the Kappa index of the simulation results was calculated as 0.9036, indicating the robustness and reliability of the DLS model.

The land use spatial pattern of Zhangye city is shown in Figure 9. It is found that the built-up area expands within a certain scale under all three scenarios, but the expansion is most pronounced under the resources consumption scenario, with large-scale concentrated and contiguous distribution occurring in northeastern Ganzhou District and southwestern Gaotai County (Figure 9b); the second is under the baseline scenario, wherein the built-up area extends mainly in the northeastern industrial park of Ganzhou District and its surrounding Linze County, with a small concentration in the southwestern Gaotai County (Figure 9a). In contrast, built-up area expansion is slowest in the green development scenario, with a relatively even and discrete spatial layout (Figure 9c). The source of the increase in built-up land area in all three scenarios is the encroachment of a large amount of cultivated land. Cultivated land is contracted on a certain scale in all three scenarios, and decreases the most in the resource consumption scenario, with a clear trend of contraction in the southwestern Gaotai County and the central parts of Minle and Shandan Counties (Figure 9b). For ecological land, grasslands and forest area expand under all three scenarios, and waters expand under the baseline scenario and green development scenario. The ecological land under the green development scenario shows the most significant expansion, and the unused land decreases considerably. The change in ecological land mainly occurs in the northern Su’nan and Gaotai Counties, as well as around the Heihe Wetland Reserve and the Dongshan Nature Reserve in Ganzhou District (Figure 9c).

Simulations of the spatial pattern of land use under different future scenarios provide reference information for the optimal allocation and management of regional land resources, sustainable economic development and ecological protection policy decisions. The simulation results show that cultivated land has a significant trend of contraction in the central part of Shandan and Minle Counties. However, these two counties mainly undertake the function of agricultural development in Zhangye. The government should pay attention to these sensitive areas of land use change, and take the relevant measures to strengthen the protection of cultivated land and basic farmland in these areas while strictly controlling the unreasonable conversion of land use. At the same time, to ensure ecological safety, the government should formulate a reasonable land resource development plan and industrial restructuring plan, considering the socioeconomic and ecological benefits of urbanization development. In the meantime, the transformation and leapfrogging of Zhangye will be promoted to achieve the goal of building “Ecological Zhangye, Green Zhangye”, so that the land use pattern of Zhangye can develop in a benign and sustainable direction that is conducive to the harmony between humans and land.

## 5. Discussion

### 5.1. Industrial Structure Adjustment and Sustainable Land Management

Combined with the current development in Zhangye city and some relevant policies, the above simulation results are reasonable. Against the background of China’s new round of Western development, the State Council, the Ministry of Natural Resources and relevant departments have issued a series of policies and measures to vigorously support Gansu Province in speeding up urbanization construction. Zhangye is in a new stage of rapidly promoting industrialization and urbanization development. With the rapid development of urbanization, the urban population increases, much of the labor force is transferred to the city, and the transfer of the industrial structure from the primary industry to the secondary and tertiary industries is becoming increasingly obvious, which leads to a continuous increase in construction land area and a continuous decrease in cultivated land area. The continuous growth of investment, consumption and exports will promote rapid economic development, and the increases in population employment and the technological progress of the labor force will promote industrial transformation, upgrading and optimization. Therefore, among the three scenarios in this study, exogenous impact factors such as investment, consumption, export, labor force and technological progress have the most significant impact on land use change in resource consumption scenarios, followed by the baseline scenario, and have the least impact on the green development scenario. According to the research on land use change in the socioeconomic system under different scenarios in the future, this paper puts forward relevant suggestions regarding urbanization development and land use planning in Zhangye city. The government should vigorously promote industrial upgrading, on the one hand, to realize the optimization of industrial structure in agriculture, and on the other hand, to actively develop the modern service industry and the tourism industry, encourage industrial sectors to strengthen technological innovation, and drive the rapid development of the regional economy. At the same time, the simulation results show that the cultivated land is reduced sharply under the resources consumption scenario. In order to ensure food security, the local government should take relevant measures to strengthen the protection of cultivated land and basic farmland, reasonably and responsibly develop and utilize land resources, and effectively supplement the cultivated land occupied by construction land.

### 5.2. Urban Development Model Based on Land Use Change

In the accelerating process of urbanization, the construction of Zhangye will manifest a great change in the pattern of land use. Zhangye is still an underdeveloped region. Economic development is still the main goal of urbanization construction. Therefore, it is easy for Zhangye to be directed by the current policy towards the one-sided pursuit of the economic benefits of the development model. The ecosystem structure of Zhangye is fragile. The natural background conditions are not superior. As an oasis city with high development intensity in the Heihe River Basin, the contradiction in Zhangye city between people and land is becoming more and more prominent in the process of rapid development. Against the strategic background of “sustainable development” and the “eco-city” concept, the rational co-ordination of ecological and economic benefits has become an unavoidable problem in the future development model of Zhangye. This study confirms that under the BAU, RCS and GDS scenarios, cultivated land is reduced to a certain extent due to the impact of the large-scale occupation of built-up areas. Since the area suitable for construction coincides with the area of high-quality cultivated land, it is easy to cause great tension between urban infrastructure construction and farmland protection. To ensure food security, relevant departments should take relevant measures to strengthen the protection of cultivated land and basic farmland, stick to the red line of cultivated land, and ensure that farmland occupied by construction will be effectively supplemented. At the same time, the ecological policy of returning farmland will also lead to the contraction of cultivated land. In the GDS scenario, the conversion of ecological land mainly occurs in Sunan County, the northern part of Gaotai County, and the Heihe Wetland in Ganzhou District and around the Dongdashan Nature Reserve. Therefore, it can also be combined with the simulation results of land use change under the GDS to explore a reasonable model for converting a large quantity of unused land into ecological land, and reduce the scale of cultivated land contraction, which can activate both the socioeconomic and ecological benefits of urbanization development.

### 5.3. Future Prospect

(1)The value accounting method of embedding land resources into the value input–output table has a great influence on the accuracy of the model simulation results. Due to the temporal and spatial differences of regional land rent and land transfer price, the value of the three major industrial land types in Zhangye city is relatively simplified by using the unified average price, so it is necessary to carry out more accurate accounting of land resources.(2)This study addresses three scenarios constrained by socioeconomic development to simulate future land use change in Zhangye city. Combined with the actual situation of the Heihe River Basin, the next step for promoting the research is to carry out scenario design under different environmental development policies, such as water resource constraint scenarios, so as to provide a more comprehensive reference for the optimal allocation of land use for sustainable development.(3)On the basis of static CGE, the model introduces capital accumulation and future trend of employment to realize a dynamic recursive model for long-term scale simulation and prediction. However, due to the partial delay of capital and investment transformation, the prediction results are unreasonable in the first few years. The model needs to be improved. Besides, we will carry out further studies on land use change and natural resource management on the watershed scale in the Heihe River Basin. Therefore, the CGELUC model will also be improved in cross-regional studies.

## 6. Conclusions

Taking Zhangye city of the Heihe River Basin, China, as an example, this paper constructs three scenarios—the baseline scenario, the resource consumption scenario and the green development scenario—to simulate land use dynamics from 2015 to 2030 by developing the CGELUC model. The DLS model is selected to dynamically simulate the spatial patterns of land use under three different future scenarios. The main conclusions are as follows:(1)Analyzing land use demand in the socioeconomic system under future scenarios, it is found that the land use demand in the three scenarios develops in the same development direction, but the degree of change is different—from the least to the greatest, they rank as the green development scenario, the baseline scenario and the resource consumption scenario. From the perspective of industrial structure, the land demand for primary industry is decreasing, the land demand for the secondary and tertiary industries is increasing, and the land demand for the tertiary industry is higher than that of the secondary industry. From the perspective of the internal industrial structure, the land demand for wheat and corn in the agricultural sector is showing a decreasing trend, but the land demand for vegetables and other agricultural sectors is increasing. For the secondary and tertiary industries, except for a few traditional industrial sectors, the land use demand for most sectors is increasing;(2)Analyzing the structure of land use change under future scenarios, it is found that land use change in the three scenarios develops in the same development direction, but the degree of change is different—from the least to the greatest, these are ranked as the green development scenario, the baseline scenario and the resource consumption scenario. The cultivated land decreases by 343 km^2^ (−8.16%), 453 km^2^ (−10.89%) and 175 km^2^ (−4.16%), respectively, under the BAU, RCS and GDS scenarios from 2015 to 2030 in Zhangye city. The built-up area increases by 29 km^2^ (8.61%), 35 km^2^ (10.39%) and 16 km^2^ (4.75%), respectively, under the BAU, RCS and GDS scenarios from 2015 to 2030. To ensure food security, relevant departments should take relevant measures to strengthen the protection of cultivated land and basic farmland, and the cultivated land occupied by construction should be effectively supplemented;(3)The simulation results of the spatial patterns of land use in Zhangye city show that the built-up area under the three scenarios undergoes a certain scale of expansion, but the expansion of the resource consumption scenario is the most significant, with contiguous distribution in the northeast of Ganzhou District and the southwest of Gaotai County. The expansion of built-up area under the green development scenario is the slowest, and the spatial patterns are relatively uniform and discrete. Cultivated land has been reduced to a certain extent under three scenarios, especially in the southwest of Gaotai County, Minle County, and central Shandan County. Ecological land is expanding, especially in the green development scenario, with a large reduction in unused land. The conversion of ecological land mainly occurs in Sunan County, the northern part of Gaotai County, and the Heihe Wetland in Ganzhou District and around Dongdashan Nature Reserve. Relevant departments should pay attention to the sensitive areas of land use change, and strictly control the unreasonable conversion of land use, giving consideration to the socioeconomic and ecological benefits of urbanization development.

## Figures and Tables

**Figure 1 ijerph-18-10501-f001:**
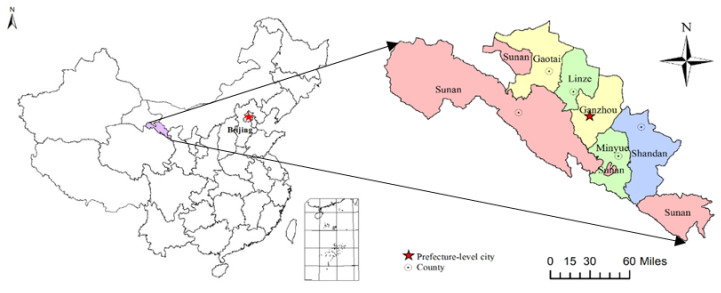
The administrative divisions of Zhangye city.

**Figure 2 ijerph-18-10501-f002:**
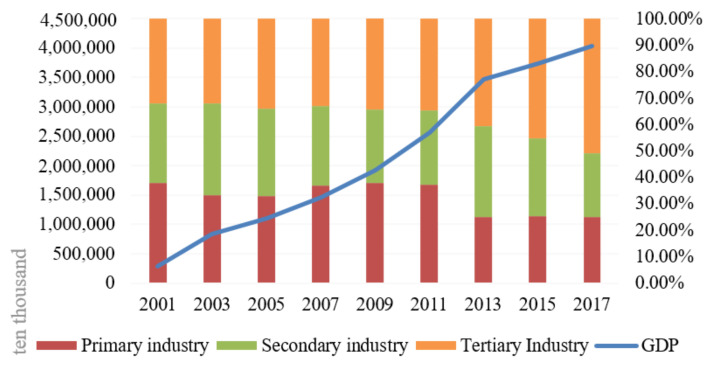
GDP and industrial structure change of Zhangye city from 2001 to 2017.

**Figure 3 ijerph-18-10501-f003:**
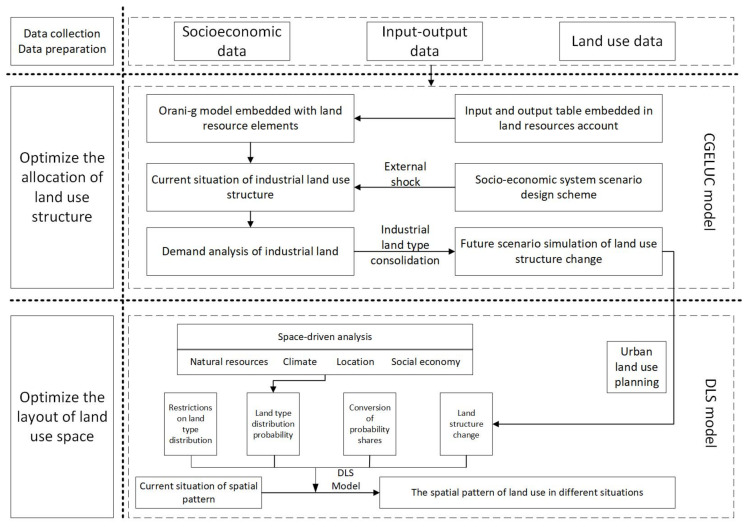
Research framework.

**Figure 4 ijerph-18-10501-f004:**
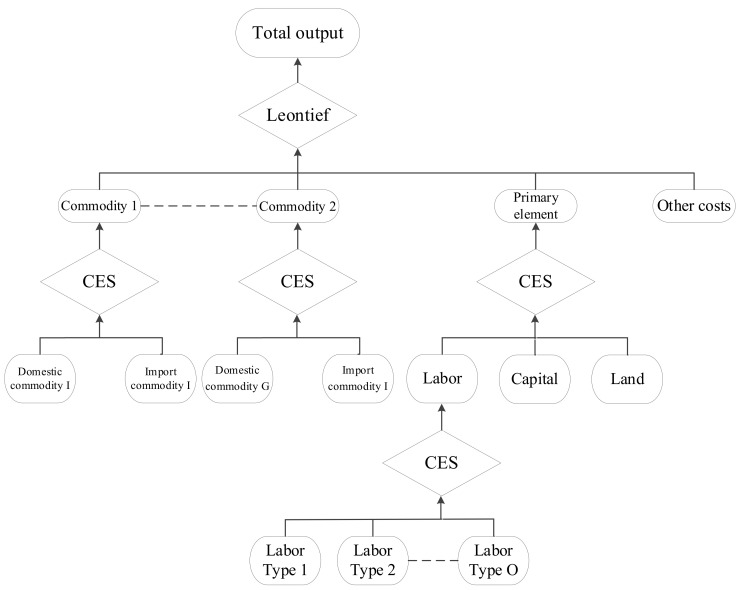
Production module of the CGELUC model.

**Figure 5 ijerph-18-10501-f005:**
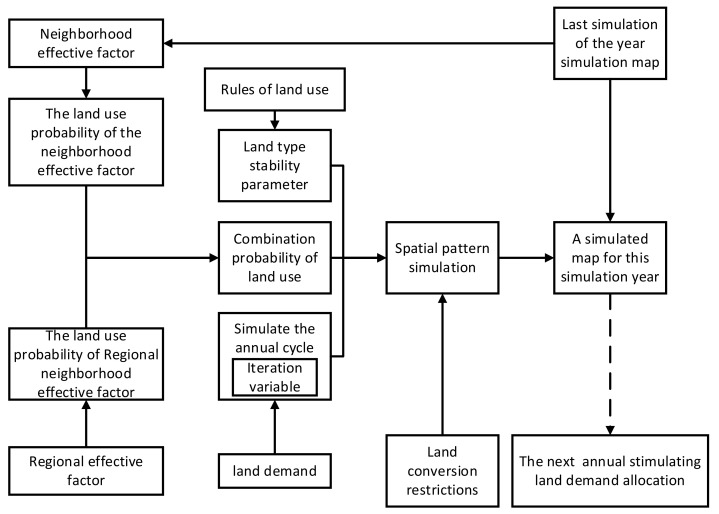
Supply and demand balance space allocation of land at the grid scale.

**Figure 6 ijerph-18-10501-f006:**
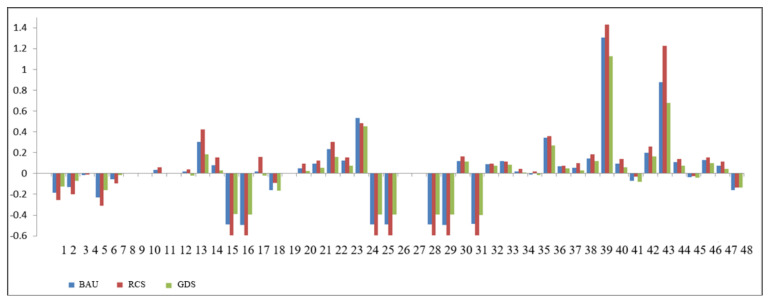
Change of land demand of each industrial sector in different scenarios. (Note: the mapping relationship between code and sectors is listed in Appendix B).

**Figure 7 ijerph-18-10501-f007:**
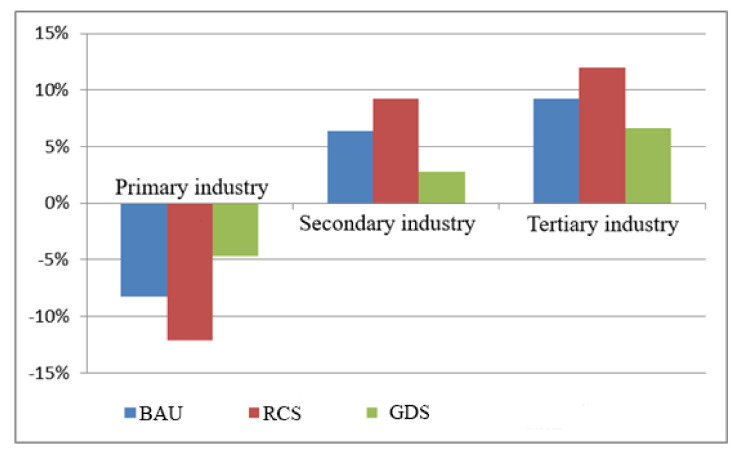
Change of three main industries’ land demand in different scenarios.

**Figure 8 ijerph-18-10501-f008:**
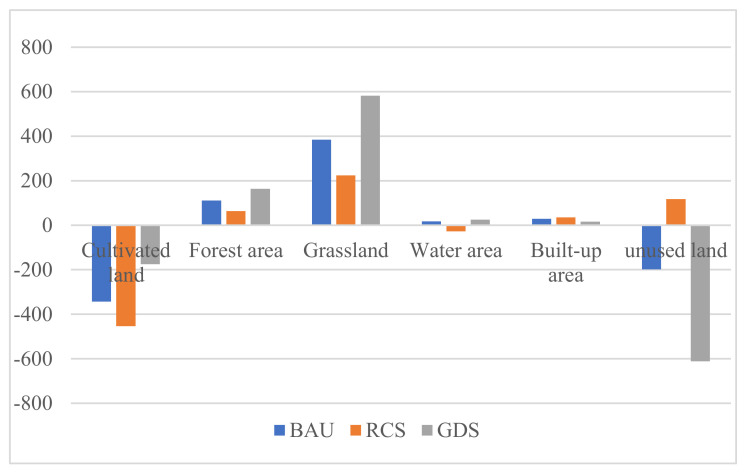
Growth of different land use types in different scenarios in Zhangye city.

**Figure 9 ijerph-18-10501-f009:**
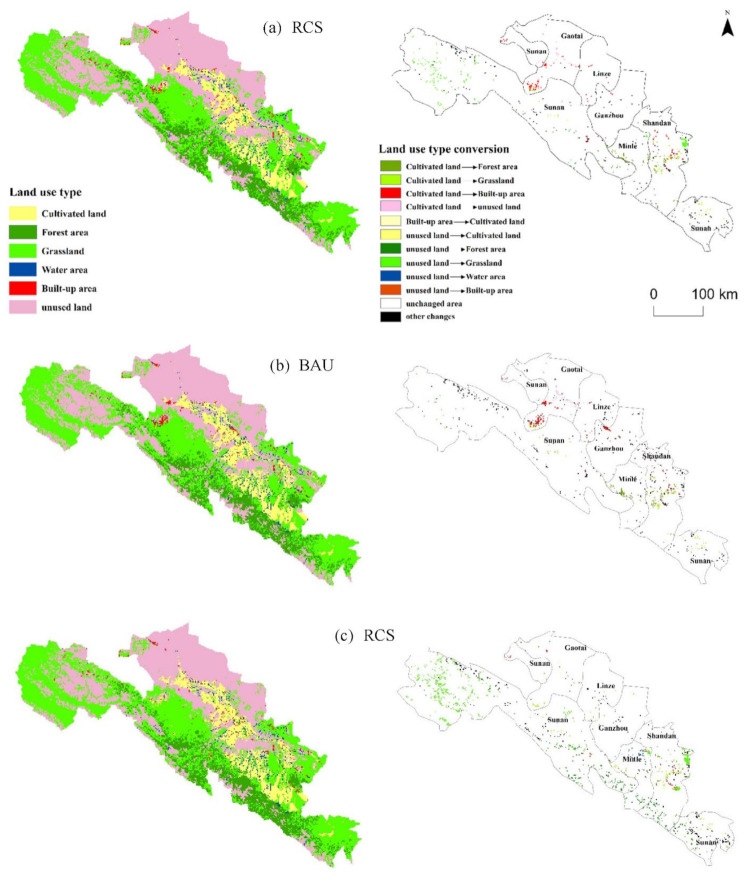
Spatial distribution of land use in Zhangye Ccty in 2030 and land use conversion during 2015–2030 under the (**a**) BAU, (**b**) RCS (**c**) and GDS scenarios.

**Table 1 ijerph-18-10501-t001:** CGE models related to land use change.

Study	Region	Model	Production Function	Number of Sectors	Base Year	Data	Policy Simulation
Alavalapati et al. [36]	Canada	CGE model	Nested Leontief/CES function	5	1990	Input–output table	Effects of restricted land use policy to prevent land degradation on economics
Olatubia et al. [37]	US	CGE model	Nested Leontief/C-D function	7	1993	Social accounting matrix	Effects of Wetland Reserve Program on agricultural economics
Lee [38]	Global	GTAP-AEZ model	Nested Leontief/CES function	57	1997	GTAP database; FAO data	Incorporate land use and land-based emissions in CGE for integrated assessment of climate change policies
Miles [39]	ARGENTINA	BioTradeLand model	Nested CES function	22	2000	Social accounting matrix	Effects of biofuel-related demand in global trade on economics
Labord and Valin [29]	Global	MIRAGE-BioF model	Nested Leontief/CES function	57	2004	GTAP database	Incorporate land use change in CGE for biofuel policies assessment
Liu et al. [40]	China	ORANI-G model	Nested Leontief/CES function	48	2012	Input–output table	Effects of agricultural water use efficiency on agricultural economics
Weng et al. [30]	China	CHEER model (Mu 2008)	Nested Leontief/CES function	8	2012	Social accounting matrix	Effects of biofuel expansion on land use change and food security
Bai et al. [41]	China	ORANI-G model	Nested Leontief/CES function	139	2012	Input–output table	Land use dynamic under climate change policies

**Table 2 ijerph-18-10501-t002:** Land use types’ areas of Zhangye city in 1990–2015 (km^2^).

Year	Cultivated Land	Forest Area	Grassland	Water Area	Built-Up Area	Unused Land
1990	3740	4281	15,136	652	300	15,359
2000	3951	4279	14,946	618	324	15,412
2005	4201	4260	14,865	619	336	15,249
2010	4204	4260	14,885	623	337	15,221
2015	4323	4303	14,733	707	413	15,032

**Table 3 ijerph-18-10501-t003:** Changing rate of shock variables in different scenarios from 2013 to 2030 (%).

Scenario	BAU	RCS	GDS
Capital growth rate	4	5.5	2.5
Labor technological progress rate	2.5	3	2.5
Investment growth rate	10.8	12.2	8.8
Resident consumption growth rate	6	7.8	4.5
Population employment growth rate	0.3	0.5	0.3
Exportation growth rate	5	6.8	3.5

## Data Availability

Not applicable.

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
