# Peer review of "Scenario-Based Analysis of Land Use Competition and Sustainable Land Development in Zhangye of the Heihe River Basin, China"

_ijerph, 2021, doi:10.3390/ijerph181910501_

Round 1

Reviewer 1 Report

I think it is a meaningful topic to simulate land-use changes in three different scenarios. In particular, I actively sympathize with the idea that land-use changes by environmental changes and economic and social factors.

Overall, it is a good paper, but I would like to present a review opinion on three matters.

1) I think it is necessary to clarify the contributions to the paper. From what I have read, the contribution of this paper is the development of the CGELUC model that developed the ORANI-G model. I want you to explain more clearly what areas have been developed and how. In particular, since the explanation of Figure 5 or the formula below it is too insufficient, it is necessary to add this. (There is also a lack of explanation for input-output data) In order to further clarify the authors' research scope, it is necessary to include a more detailed analysis of previous studies. Currently, related works are analyzed only by explanation in the text, and it would be better to specify them using a table. Through this, it will be more clearly compared to what models or variables some papers have written and where the authors' papers are positioned.

2) I hope that the basis for the exogenous variables specified in Table 2 will be clarified. If you can't explain all the variables, you'd better explain why even one or two of them came out. This is very important to help readers understand.

3) It is necessary to add the following matters in the visual notation of the paper. In the case of Fig. 10, the DLS and remote sensing results simulated in this study were visualized by dividing them into a and b, but the figure itself is so similar that it looks like copy & paste. Of course, I think that the Kappa coefficient is high enough, but in order to compare the difference (a) and (b), it is good just to add the RS image. In addition, in Figure 11, it would be better to mark the changed part over time and write it in a way that explains the marked part in the text. Currently, the change situation is not expressed correctly as the land-use situation for large areas is captured on a small scale. In addition, it is written in the text that the grid-scale is used when applying the DLS model, so it is better to indicate the exact spatial unit.

Author Response

Response to Comments from Reviewer #1

Reviewer#1

I think it is a meaningful topic to simulate land-use changes in three different scenarios. In particular, I actively sympathize with the idea that land-use changes by environmental changes and economic and social factors.

Overall, it is a good paper, but I would like to present a review opinion on three matters.

Authors:

Thanks for your comments. We have revised our manuscript accordingly using tracking changes.

Reviewer#1 (1):

1) I think it is necessary to clarify the contributions to the paper. From what I have read, the contribution of this paper is the development of the CGELUC model that developed the ORANI-G model. I want you to explain more clearly what areas have been developed and how. In particular, since the explanation of Figure 5 or the formula below it is too insufficient, it is necessary to add this. (There is also a lack of explanation for input-output data) In order to further clarify the authors' research scope, it is necessary to include a more detailed analysis of previous studies. Currently, related works are analyzed only by explanation in the text, and it would be better to specify them using a table. Through this, it will be more clearly compared to what models or variables some papers have written and where the authors' papers are positioned.

Authors:

Thanks for your comments and suggestions. We have added one paragraph with a table, which compared different models and its application to explain the advances and limitations of CGE models related to land use change in the section of Introduction. We also added more introduction about ORANI-G model and what we improve in the model in the section of Methodology. In terms of input-output data, we used the Zhangye Input-Output table obtained from the Bureau of Statistics of Zhangye (2013) and provide a more detailed explanation on how to embed the land resources account into input-output table in 3.3.3.

Reviewer#1 (2):

2) I hope that the basis for the exogenous variables specified in Table 2 will be clarified. If you can't explain all the variables, you'd better explain why even one or two of them came out. This is very important to help readers understand.

Authors:

Thanks for your comments and suggestions. We added explanation accordingly in section 3.4.

Reviewer#1 (3):

3) It is necessary to add the following matters in the visual notation of the paper. In the case of Fig. 10, the DLS and remote sensing results simulated in this study were visualized by dividing them into a and b, but the figure itself is so similar that it looks like copy & paste. Of course, I think that the Kappa coefficient is high enough, but in order to compare the difference (a) and (b), it is good just to add the RS image. In addition, in Figure 11, it would be better to mark the changed part over time and write it in a way that explains the marked part in the text. Currently, the change situation is not expressed correctly as the land-use situation for large areas is captured on a small scale. In addition, it is written in the text that the grid-scale is used when applying the DLS model, so it is better to indicate the exact spatial unit.

Authors:

Thanks for your comments and suggestion. We revised the section 4.3 according to your comments, added the figures to show the spatial changes of land use, and added explanation and description that explains the spatial changes of land use in the text.

Reviewer 2 Report

First of all, congratulations for the work.

I have found small errors that should be modified. It is advisable to add line numbers to make it easier to indicate improvements.

Is the CGE model the same as the CGELUC model? If yes, unify nomenclature.

Enter full name of CA model.

Change Fig.3. to Fig. 3.

Add figure 4 to the text (introduction of the figure)

I am having trouble seeing the numbering of the equations (1) to (4).

In 3.3.3 Change 13.6 6Yuan to 13.66 Yuan.

The legends of figures 10 and 11 are illegible. Try to make it bigger.

Use the same format in the tables.

With these changes I believe the article will improve. Best of luck.

Author Response

Reviewer#2:

First of all, congratulations for the work.

I have found small errors that should be modified. It is advisable to add line numbers to make it easier to indicate improvements.

Is the CGE model the same as the CGELUC model? If yes, unify nomenclature.

Enter full name of CA model.

Change Fig.3. to Fig. 3.

Add figure 4 to the text (introduction of the figure)

I am having trouble seeing the numbering of the equations (1) to (4).

In 3.3.3 Change 13.6 6Yuan to 13.66 Yuan.

The legends of figures 10 and 11 are illegible. Try to make it bigger.

Use the same format in the tables.

With these changes I believe the article will improve. Best of luck.

Authors:

Thanks for your comments. We have revised our manuscript accordingly using tracking changes.

We added the full name of CGE model in the Introduction. CGE model is computable general equilibrium model, while CGELUC model is a computable general equilibrium of land use change (CGELUC) model. We also added full name of CA model and corrected other errors accordingly.
